# Development and Validation of an Explainable Machine Learning-Based Model for Predicting the Interval Growth of Pulmonary Subsolid Nodules: A Prospective Multicenter Cohort Study

Zhedong Zhang[1,2,3#], Chao Sun[1,4#], Lixin Zhou[1,2], Hao Li[1,2], Xianjun Min[5], Jiabao Liu[6], Zhitao Wang[7], Dawei Wang[8], Nan Hong[1,4*], Fan Yang[1,2*], Xiao Li[1,2*]

[1] Department of Thoracic Surgery, Peking University People's Hospital, Beijing 100044, PR China.

[2] Thoracic Oncology Institute, Peking University People's Hospital, Beijing 100044, PR China.

[3] Department of Thoracic Surgery, Second Affiliated Hospital, Zhejiang University School of Medicine, Zhejiang University, Hangzhou 310009, PR China.

[4] Department of Radiology, Peking University People's Hospital, Shijiazhuang 050051, PR China.

[5] Department of Thoracic Surgery, AMHT Group Aerospace 731 Hospital, Beijing 100044, PR China.

[6] Department of Thoracic Surgery, Shijiazhuang People's Hospital, Beijing 100044, PR China.

[7] Yijian Dian Health Management Center, Peking University Health Science Center, Beijing 100044, PR China.

[8] Institute of Advanced Research, Infervision Medical Technology Co., Ltd, Beijing 100025, PR China.

Zhedong Zhang and Chao Sun contributed equally to this work and shared the first authorship.

* Corresponding authors at: Peking University People's Hospital, No. 11 Xizhimen South Street, Xicheng District, Beijing 100044, China (Nan. H). Peking University People's Hospital, No. 11 Xizhimen South Street, Xicheng District, Beijing 100044,

29   China (F. Yang). Peking University People's Hospital, No. 11 Xizhimen South Street,

30   Xicheng District, Beijing 100044, China (X. Li).

31   E-mail address: dr.lixiao@163.com (X. Li)

32

33   **Abstract**

34   **Objectives:** In this multicenter study, we aimed to develop and validate a predictive model for

35   pulmonary subsolid nodules (SSN) growth at different time intervals by machine learning (ML)

36   based CT radiomics methods. This model is intended to guide personalized follow-up strategies in

37   clinical practice.

38   **Methods:** A total of 642 patients with 717 SSNs who underwent long-term follow-up were

39   retrospectively collected from three medical centers. Patients were categorized into growth and non-

40   growth groups based on the growth status of presented SSNs within 2 or 5 years, and they were

41   randomly divided into training and internal testing sets at an 8:2 ratio. Predictive models were

42   developed using the optimal ML algorithms for clinical, radiomics, and clinical-radiomics fusion

43   models to assess the risk of SSN growth over different timeframes. An independent external test set

44   was established by including another 95 patients with 105 SSNs from a health examination center.

45   Multiple assessment indices, including the area under the receiver-operating-characteristic curve

46   (AUC), were utilized to assess and compare predictive performance. Furthermore, the SHapley

47   Additive exPlanation (SHAP) method was employed to rank the importance of features and

48   elucidate the rationale behind the final model.

49   **Results:** The extreme gradient boosting (XGBoost) and light gradient boosting machine (Light

50   GBM) model performed best in discriminative ability among the 8 ML models. For the prediction

51   of within-2-year growth, the clinical, radiomics, and clinical-radiomics fusion models developed

52   using the optimal ML algorithms achieved the AUC of 0.823 (95% CI: 0.745-0.906), 0.889 (95%

53   CI: 0.823-0.943), and 0.911 (95% CI: 0.858-0.955) on the internal testing set, and the AUC of 0.712

54   (95% CI: 0.610-0.815), 0.734 (95% CI: 0.616-0.830), and 0.734 (95% CI: 0.623-0.835) on the

55   external testing set. In 5-year growth prediction task, the three models achieved AUCs of 0.796 (95%

56   CI: 0.708-0.884), 0.838 (95% CI: 0.759-0.905), and 0.849 (95% CI: 0.772-0.913) on the internal

57   testing set and AUCs of 0.672 (95% CI: 0.550-0.795), 0.773 (95% CI: 0.657-0.880), and 0.776 (95%

CI: 0.652-0.882) on the external testing set. Furthermore, these insights have been translated into a streamlined clinical management framework, enhancing its utility within clinical settings.

**Conclusions:** The interpretable machine learning model we developed based on multicenter longitudinal follow-up data for SSN has been successfully developed to accurately predict changes in SSN over 2 years and used for the first time to guide 5-year long-term follow-up.

**Keywords:** Subsolid nodules, Natural course, Lung adenocarcinoma, Radiomics, Machine learning.

## 1. Introduction

Lung cancer is the most common cancer globally, and with the widespread use of Low-Dose Computed Tomography (LDCT), the detection rate of pulmonary subsolid nodules (SSN) has significantly increased[1]. Previous research has indicated that the appearance of growth during follow-up strongly suggests malignancy in SSN[2, 3]. Recent studies have shown that SSNs that persist and remain stable for over five years still require continued monitoring, and extended follow-up beyond five years may reveal more cases of lung cancer[4-7].

Due to differences in the biological characteristics and prognosis of pure ground-glass nodules (pGGN) and part-solid nodules (PSN), major guidelines manage these two types separately. However, there is considerable controversy in determining the presence of solid components within SSN and measuring the solid components[8]. Some PSNs have solid components visible in both the lung and mediastinal windows (real part-solid nodules, rPSN). In contrast, others have solid components only visible in the lung window (heterogeneous ground-glass nodules, hGGN). Previous research has reported that the average time for hGGN to develop into rPSN is 2.1 years[9], and rPSN has poorer clinical pathological results and prognosis than hGGN[10]. Previous studies on the natural course of SSN have primarily focused on comparing differences between pGGN and PSN based on lung window classification[11-13]. However, these studies have not provided sufficient information on the natural course of solid components under different window settings. Current guidelines for SSN management have several limitations. Therefore, establishing personalized SSN management methods and developing appropriate follow-up strategies hold significant clinical importance.

Radiomics, as a non-invasive method, can extract numerous features from Computed Tomography (CT) scan images through high-throughput computation, transform them into

comprehensive quantifiable data, and develop models to non-invasively predict various phenotypic features of lesions[14, 15]. Numerous radiomics studies have established predictive models to enhance the accuracy of diagnosing benign and malignant nodules, assessing the degree of infiltration, and predicting histological subtypes and prognosis in lung cancer patients[16]. However, due to limitations in the number of cases and follow-up time, how to dynamically track nodules and predict the growth patterns of SSNs using radiomics remains to be further explored [17, 18].

Based on these scientific questions, this study selected SSN patients from multiple centers with long-term follow-up as the research subjects, including those with growth at two years and five years of follow-up and those with sustained stability. Through ML modeling, clinical and radiomics features of patients were used to establish clinical-radiomics fusion models to predict the growth status of SSNs within two years and five years, optimizing individualized follow-up strategies.

## 2. Material and Methods

*2.1 Study Design and Inclusion Criteria*

This study is a multi-center retrospective cohort study based on the STORBE guidelines. It includes adult individuals aged 18 years or older who underwent chest CT scans for any reason at three tertiary comprehensive medical centers from November 2007 to August 2021, regardless of their smoking history. They were randomly divided into a training set and an internal testing set in an 8:2 ratio. In addition, patients from a medical examination center in Beijing, China, who underwent chest CT examinations between January 2017 and November 2021 were included as an independent external testing set. The inclusion and exclusion criteria were identical to those of the derivation cohort. **(Detailed information on participating hospitals was listed in Supplementary Materials)**. Clinical data, such as smoking history, previous history of pulmonary conditions and malignancies, and chest CT scans of enrolled patients, were all retrospectively collected for modeling. Notably, our study had two key components **(Figure 1)**. First, we constructed predictive models for growth at different time intervals. We pretested the probability of nodule growth using eight typical machine learning algorithms and selected the best-performing algorithm for further optimization. Secondly, the model underwent external validation in an independent external SSN cohort to assess its performance.

The inclusion criteria for SSN were as follows: (1) confirmation of SSN persisting for at least

118  six months after the initial chest CT examination; (2) SSN with a maximum long-axis diameter of

119  ≤3 cm; (3) evaluation of each SSN using original Digital Imaging and Communications in Medicine

120  (DICOM) format files from chest CT with a slice thickness of ≤1.5 mm; (4) a minimum follow-up

121  time of at least two years or a follow-up time of less than two years but with documented nodule

122  growth during the follow-up period. Exclusion criteria were: (1) inability to obtain detailed clinical

123  data for patients with SSN; (2) a decrease in the maximum long-axis diameter of the SSN by ≥2 mm

124  during the follow-up period.

125  Non-smokers were defined as individuals who had never smoked in their lifetime or had

126  smoked fewer than 100 cigarettes. Individuals with data on smoking status but without data on the

127  quantity of cigarettes smoked were included. Exclusion criteria included participants with a history

128  of lung cancer at the baseline screening and those with unknown smoking status history.

129  This study obtained approval from the Institutional Review Boards (IRB) of three tertiary

130  comprehensive medical centers (No. 2022PHB031-001, No. 2022-0601-01, and No. 2022002). The

131  IRB waived the requirement for written informed consent from the participants.

132

133  *2.2 Growth Definition and SSNs Categorization*

134  SSN growth is defined as occurring during follow-up[19]: (1) an increase in the maximum

135  diameter by ≥2mm; (2) an increase in the solid component of PSN (including hGGN and rPSN) by

136  ≥2mm; (3) the appearance of any diameter of a solid component in pGGN under the lung window.

137  SSN reduction is a decrease in the maximum diameter of the solid component by ≥2mm during

138  follow-up[20]. SSN stability is defined as not meeting the criteria for growth or reduction during

139  follow-up. SSN is categorized into pGGN, hGGN, and rPSN based on the radiological features of

140  lung and mediastinal windows in chest CT.

141  Given that the 2-year and 5-year progression-free survival serve as important prognosis

142  indicators for patients with cancer[4, 5, 12], we accordingly studied the growth risks of SSN within 2-

143  year and 5-year periods. In particular, the enrolled patients were categorized according to the status

144  of SSN (growth or stable) within two years and five years, respectively. SSNs were labeled as

145  positive if they grew within the 2 or 5-year follow-up period and assigned as negative if they

146  remained stable during the 2 or 5-year follow-up period, regardless of subsequent growth afterward.

*2.3 Clinical and Radiographic Data Collection*

We collected primary clinical, surgical, and pathological data of SSN patients included in the study. This had demographic information, surgical procedures, and the pathological diagnosis of SSNs in patients who underwent resection surgery. A chief thoracic surgeon or higher decided for the need for invasive tests to obtain pathological results for SSNs. The pathological types of SSNs were categorized as benign, lung adenocarcinoma (LUAD) (invasive adenocarcinoma, IAC, and minimally invasive adenocarcinoma, MIA), and their epithelial precursor lesions (adenocarcinoma in situ, AIS and atypical adenomatous hyperplasia, AAH).

We gathered imaging data for the included SSNs, encompassing both CT quantitative features and non-quantitative features. The detailed information regarding CT acquisition and parameters is described in **Figure S1**. This data had the maximum diameter of SSNs, the diameter of the solid component under the lung window (LW) and mediastinal window (MW), the Lung Window-Consolidation Tumor Ratio (LW-CTR), the Mediastinal Window-Consolidation Tumor Ratio (MW-CTR), and the number of multifocal SSNs. We also collected information on SSN types, lobulation signs, air bronchograms, vascular signs, pleural tag signs, and others. In discrepancies, a consensus was reached through discussion with a third radiologist.

*2.4 Nodules Segmentation and Feature Selection*

A radiology-trained thoracic surgeon (Reader 1, with ten years of chest imaging experience) and a radiologist (Reader 2, with 15 years of experience in reading chest imaging) independently performed layer-by-layer SSN segmentation based on radiological examination reports. The ROI of each nodule layer was manually drawn based on the image using 3D Slicer (version 5.2.1) **(Figure S2)**. If there were multiple SSNs, the largest one was selected for analysis. To ensure the stability of feature extraction, the consistency of radiomic features was assessed between different observers and by the same observer at other times. Three months later, 30 nodules were randomly selected and segmented again by the above two readers. Two readers were blinded to clinical characteristics, SSN growth, and histopathology.

A voxel size standardization of 1mm was applied along the x, y, and z axes. Pyradiomics (version 3.0.1) was used to extract SSN radiomic features, resulting in a total of 1454 radiomic features, including first-order features, Gray Level Cooccurrence Matrix (GLCM) features, Gray

178     Level Dependence Matrix (GLDM) features, Gray Level Run Length Matrix (GLRLM) features,

179     Gray Level Size Zone Matrix (GLSZM) features, Neighboring Gray Tone Difference Matrix

180     (NGTDM) features, and shape-based features. Detailed information on extracted features was

181     summarized in **Supplementary Table 1**.

182         Clinical features, including clinical characteristics and conventional imaging features, were

183     selected for subsequent model construction through univariate and multivariate logistic regression.

184

185     *2.5 Model Construction, Selection, and Validation*

186         Eight machine learning models, including Logistic Regression (LR), Random Forest (RF),

187     Support Vector Machine (SVM), Naive Bayes (NB), Extreme Gradient Boosting (XGBoost), Light

188     Gradient Boosting Machine (Light GBM), K-Nearest Neighbor (KNN), and Multilayer Perceptron

189     (MLP), were trained with selected clinical and radiomic features to build three sets of models:

190     clinical models, radiomic models, and clinical-radiomic fusion models. The models with the highest

191     diagnostic performance parameters were selected using repeated three-fold cross-validation on the

192     training dataset. The model with the best AUC calculated on the internal testing dataset was used as

193     the final model for application on the external testing dataset. Discrimination was quantified using

194     the area under the curve (AUC). Several commonly used evaluation indexes, such as the area under

195     the receiver-operating-characteristic (ROC) curve (AUC), sensitivity, specificity, positive predictive

196     value (PPV), negative predictive value (NPV), accuracy, and F1 score, were used to evaluate the

197     reliability of these models. Predictive accuracy was assessed using calibration curves and confusion

198     matrices. Shapley Additive exPlanations (SHAP) was used to visualize the correlations between

199     variables and SSN growth.

200

201     *2.6 Follow up*

202         The growth interval refers to the time from the baseline chest CT scan to the subsequent

203     follow-up CT scan, during which the same SSN met the criteria for growth. The total

204     observation time is the interval between the baseline and final CT scans for the same SSN or

205     between the baseline CT scan and the SSN's last intervention. The follow-up CT intervals were

206     determined by specialized clinical thoracic surgeons based on patient and SSN radiological

207     features, following guidelines.

208     All patients were followed up via phone or outpatient visits, and the outcomes of the nodules

209     were recorded. The final follow-up deadline was December 2022.

210

211     *2.7 Statistical analysis*

212     Statistical analysis was conducted using SPSS (version 26), R software (version 3.6.2), and

213     Python (version 3.11). For data following a normal distribution, values were presented as mean

214     ± standard deviation (SD), and intergroup comparisons were performed using independent

215     sample t-tests. Data with non-normal distribution were described as median [interquartile range,

216     IQR] and analyzed using the Mann-Whitney U test. Categorical variables were presented as

217     frequencies and percentages, and intergroup comparisons were made using the Chi-square test

218     or Fisher's exact test when appropriate. The DeLong test was employed to compare different

219     ROC curves. Feature selection for radiomics and model construction was made using Python's

220     "scikit-learn" machine learning framework. ROC curves and confusion matrices were

221     generated using Python's "Matplotlib" library. A significance level of $p < 0.05$ was considered

222     for all tests.

223

224     **3. Results**

225     *3.1 Baseline Clinical Characteristics of the Patients and Radiologic Features of Included SSNs*

226     This study included 642 patients with 717 SSNs from three different hospitals and 99

227     patients with 105 SSNs from one medical examination center. Two-year and five-year growth

228     prediction models were established based on whether SSNs grew within 2 and 5 years,

229     respectively. Clinical baseline characteristics of all patients and radiological characteristics of

230     all SSNs are summarized in **Table 1** and **Table 2**.

231     All patients and SSNs from three tertiary comprehensive medical centers were randomly

232     divided into training and internal test sets in an 8:2 ratio. Patient information and SSN features

233     for the two datasets are compared in **Table S2 and Table S3.** There were no significant

234     differences in the pathological characteristics and surgical methods of SSNs that underwent

235     surgical resection in the training and internal testing sets; details are provided in **Table S4**.

236

237     *3.2 Clinical Feature selection, model development, and performance comparison*

238     The univariate and multivariate analyses revealed that gender, SSN type, vascular sign,

239 and initial maximum diameter were independent risk factors for the different-year prediction

240 models. In the 5-year prediction model, the vacuole sign was a newly discovered factor **(Table**

241 **S5)**. The clinical models established based on the optimal machine learning algorithm,

242 XGBoost, achieved an AUC of 0.823 (95% CI: 0.745-0.906) and 0.796 (95% CI: 0.708-0.884)

243 in the internal testing cohort for the 2-year and 5-year predictions, respectively (Comparison of

244 Clinical Models Established by Different Machine Learning Algorithms and ROC Curves are

245 presented in Supplementary Material **Figure S3** and **Table S6**).

246

247 *3.3 Radiomics Feature selection, model development, and performance comparison*

248  Among the 1,454 radiomic features, redundant features (ICC < 0.75 and PCC > 0.9) were

249 first removed, resulting in 271 and 259 remaining features for the 2-year and 5-year models,

250 respectively. The final set of 10 radiomic features was generated through recursive feature

251 elimination and cross-validation **(Table S7)**. For the 2-year model, the radiomic model was

252 developed using the optimal machine learning algorithm LightGBM, achieving an AUC of

253 0.889 (95% CI: 0.823-0.943) in the internal testing cohort. For the 5-year model, the radiomic

254 model was developed using the optimal machine learning algorithm XGBoost, achieving an

255 AUC of 0.838 (95% CI: 0.759-0.905) in the internal testing cohort **(Figure S4 and Table S8).**

256

257 *3.4 Development of fusion radiomics model and performance comparison*

258  Incorporating the selected clinical features and radiomic features, a clinical-radiomic

259 fusion model was developed using the mentioned eight machine learning algorithms (**Figure**

260 **S5 and Table S9**). The results show that for the 2-year prediction model, the LightGBM

261 algorithm with its specific parameters performed the best on the internal testing dataset,

262 achieving an AUC of 0.911 (95% CI: 0.858-0.955). For the 5-year prediction model, the

263 XGBoost algorithm with its specific parameters performed the best on the internal testing

264 dataset, resulting in an AUC of 0.849 (95% CI: 0.772-0.913).

265

266 *3.5 Identification of the Final Model and External Validation*

267  The optimal algorithm with the highest AUC in the internal testing set was selected to

268 build the model, and its predictive performance was compared. **Table 3** displays the predictive

269 performance of the three groups of prediction models, and ROC curves are presented in **Figure**

270 **2**. The addition of SHAP allows for interpretative analysis of the fused radiomic model by

271 visualizing the specific impact of each variable on the prediction of SSN growth **(Figure 3).** In

272 the 2-year prediction model, the fusion model produced the highest AUC in the internal testing

273 cohort. Both the fusion model and radiomics model had higher AUC values compared to the

274 clinical model (DeLong test p < 0.05), with no significant difference between the fusion model

275 and radiomics model (DeLong test p > 0.05). In the internal testing set of the 5-year prediction

276 model, even though there were no statistically significant differences among the three models,

277 the fusion model achieved the highest AUC. Simultaneously, the radiomic and fusion models

278 exhibit higher accuracy and sensitivity.

279     In the external testing set of the 2-year prediction model, the AUC values for the clinical

280 model, radiomics model, and clinical-radiomics fusion model were 0.712 (95% CI: 0.610-

281 0.815), 0.734 (95% CI: 0.616-0.83), and 0.734 (95% CI: 0.623-0.835), respectively. In the

282 external testing set of the 5-year prediction model, the AUC values for the three groups were

283 0.672 (95% CI: 0.550-0.795), 0.773 (95% CI: 0.657-0.880), and 0.776 (95% CI: 0.652-0.882),

284 respectively **(Figure 4)**. The metrics for evaluating the reliability of these models are outlined

285 in **Table 4**. The confusion matrices for each model are displayed in **Figures S6 and S7**. This

286 suggests that the combined radiomics model exhibits good stability and reproducibility.

287

288 *3.6 Follow-up management framework for clinical utility*

289     Based on the research outcomes, we propose a tailored follow-up management framework

290 for SSN, outlined in **Figure 5**. Firstly, for patients identified with SSN via thoracic thin-section

291 CT, clinical physicians assess whether further examination, surgery, no follow-up, or regular

292 follow-up is necessary. If regular follow-up is deemed necessary, the SSN is subjected to growth

293 prediction models for evaluation. If the 2-year growth prediction model indicates a high risk of

294 growth, clinical physicians may propose high-risk management recommendations after a

295 thorough review. This involves extensive discussion with the patient to determine whether to

296 continue follow-up with a shortened interval, undergo further examination, or proceed with

297 surgery. The SSN is further evaluated using a 5-year growth prediction model if the growth risk

298 is low. Suppose the 5-year growth prediction model indicates a high risk of SSN growth. In that

case, appropriate management recommendations are proposed after a thorough evaluation, possibly shortening the follow-up interval for continued monitoring and considering further examination if necessary. If the growth risk is low, low-risk management recommendations are provided, suggesting regular follow-up for the patient and possibly extending the follow-up interval appropriately.

**4. Discussion**

In clinical practice, determining the nature of nodules and establishing follow-up duration is crucial for long-term persist and stable SSN. In this study, based on multicenter, long-term follow-up cases, we constructed a growth prediction model using machine learning methods to predict the growth status of SSN at 2 and 5 years, aiming to assist in the standardized management of pulmonary nodules. This clinical prediction model demonstrates excellent diagnostic performance and has been validated using independent external data.

Pulmonary SSN growth modes were categorized into five patterns based on consecutive follow-up CT scans, including linear, rapidly accelerating, slow accelerating, slow, and rapid growth while pointing out that the likelihood of malignant radiological features of nodules increases with prolonged follow-up[18]. Therefore, index models based on volume doubling time (VDT) may not be suitable for evaluating every nodule encountered in clinical practice. Studies have shown that after two years or more of stability, the probability of SSN experiencing growth is only 5%[21]. Additionally, within three years, 26.9% of SSNs showed growth; among those stable for three years and followed up to five years, 6.7% demonstrated growth[22, 23]. Taking into account the aforementioned concerns, this study categorized subjects into groups based on their growth within two years and five years respectively. Utilizing predictive modeling, it aimed to assess growth risks and devise tailored follow-up strategies for individuals across different growth durations.

The progression of SSNs represents a complex and dynamic process. As anticipated, this study demonstrates that gender, larger size on initial CT imaging, and the presence of solid components are independent risk factors for SSN progression, consistent with previous research [5, 6, 9, 23, 24]. Introducing the radiographic characteristics of CT MW images into classification, SSNs are divided into pGGN, hGGN, and rPSN. It was found that hGGN and rPSN have

329    differences in growth patterns[9, 25]. Further analysis combining large-panel targeted sequencing

330    confirmed at the genomic level that only hGGNs with solid components on LW are intermediate

331    subtypes of PSNs. Meanwhile, the genomic structure of hGGNs is closer to pGGNs[26]. The

332    assessment of nodule growth should not only depend on size changes but may also be

333    influenced by morphological features. Vascular signs are defined as either vessel traversing

334    through the lesion or vascular thickening and tortuosity around the lesion, indicating a higher

335    demand for blood supply and often indicative of malignancy[27]. The relationship between

336    vessels and SSNs can be classified into three types: Type I (intact vessels passing through or

337    traversing the SSN without tiny branches), Type II (intact vessels passing through the SSN

338    without tiny branches), and Type III (distorted vessels within the SSN are wider or tortuous).

339    Type II and Type III are more likely to be associated with malignancy than Type I. Hence, the

340    relationship between vessels and lesions might predict SSN progression[28]. Similarly, vacuole

341    sign has also been established as predictive factors for the growth of SSNs in previous studies

342    [24].

343        The most commonly used method currently is to select radiomic features through LASSO

344    regression and then estimate each radiomic feature using logistic regression algorithms to

345    calculate a Rad-score for predicting SSN growth[12, 17]. In this study, we applied dimensionality

346    reduction to 1454 radiomic features using the REF method and built models using the ten

347    optimal features associated with SSN growth in 2 and 5 years, which outperformed previous

348    reports. We also found that the most influential feature in the 2-year radiomics model was

349    glszm_HighGrayLevelZoneEmphasis_original, which assists in quantifying regions of high

350    brightness in the image[29]. In the 5-year radiomics model, the most influential feature was

351    glrlm_LongRunEmphasis_wavelet-LLH, which measures the frequency and intensity of

352    continuous appearances of pixels with the same grayscale values in the image[30].

353        Previous studies have utilized LR on single-center retrospective data to build clinical-

354    radiomics nomograms for predicting 2-year growth of uncertain pulmonary nodules,

355    emphasizing the importance of combining clinical and radiomic features in predicting nodule

356    growth. However, the inclusion of nodules confirmed by histology and a high proportion of

357    malignant nodules in these studies led to an overestimation of the diagnostic performance of

358    the model[17]. Similarly, Chen et al. established clinical-radiomics nomograms for predicting

359    the growth of SSN beyond two years based on single-center data, with the combined model

360    significantly outperforming the clinical model but no significant difference between the

361    combined model and the radiomics model. This study did not incorporate patients' clinical

362    information into the analysis. [12]. Yang et al. developed several machine-learning models to

363    predict whether lung nodules would grow within one year. They found that a LR model

364    combining age and radiomic features performed the best (with an AUC of 0.87 in the training

365    set and 0.82 in the validation set)[31]. However, the studies mentioned above all suffer from

366    relatively small sample sizes and suboptimal model performance. With improved

367    computational capabilities and storage space availability, machine learning algorithms can

368    analyze more complex data and provide real-time output[32, 33]. XGBoost has recently become

369    a popular algorithm, gaining recognition in various machine-learning competitions[34].

370    LightGBM, compared to XGBoost, has the advantage of faster training speed and lower

371    memory usage[35]. Both machine learning methods outperform traditional linear models in terms

372    of predictive accuracy.

373    Building upon the aforementioned studies, we have conducted exploratory analyses and

374    established, for the first time, a clinical model to predict whether SSNs will grow within 5 years

375    based on multi-center, long-term follow-up data. The results demonstrate that in the internal

376    testing set, the AUC values for the clinical model, radiomic model, and clinical-radiomic fusion

377    model were 0.796 (95% CI: 0.708-0.884), 0.838 (95% CI: 0.759-0.905), and 0.849 (95% CI:

378    0.772-0.913), respectively. According to the DeLong test, there was no significant difference

379    among the three models, suggesting that the clinical and combined models have equivalent

380    efficacy in predicting whether SSNs will grow within 5 years. However, the fusion model

381    exhibited higher accuracy (0.730 vs. 0.680), sensitivity (0.767 vs. 0.517), F1 score (0.7773 vs.

382    0.660), and AUPRC (0.909 vs. 0.859) compared to the clinical model. Therefore, future

383    investigations using prospective data from long-term follow-up can further explore the

384    predictive value of the combined model in predicting SSN growth within 5 years.

385    Mainstream medical societies have issued guidelines for managing pulmonary nodules,

386    but these guidelines differ in scope and emphasis. They primarily focus on factors such as

387    malignancy probability thresholds, follow-up schedules based on imaging features, malignancy

388    risk calculators, and the use of VDT [36-41]. To address the complexity of managing pulmonary

389  nodules, we utilized the SHAP method to provide insights into the inner workings of machine

390  learning models. We developed personalized prediction models based on clinical and imaging

391  data to enhance clinician acceptance and decision-making. Despite the initial success, when

392  extending the model to health examination centers, there was a decrease in diagnostic

393  performance. However, the model still exhibited stability and reliability. Therefore, in the

394  clinical setting, clinicians should view these models as valuable tools but also consider

395  individual patient factors and exercise flexibility in decision-making to ensure personalized and

396  accurate diagnosis and treatment.

397      However, this study has some limitations. First, it is a multi-center retrospective

398  observational study, and it may have selection bias and temporal bias. Clinical practitioners

399  influence variations in follow-up intervals and duration at different medical centers. Second,

400  the study only included an Asian population, which may have significant demographic

401  differences compared to a Caucasian population. Therefore, clinical prediction models cannot

402  be assessed for patients of different ethnicities, and further validation of the results is needed

403  through international multi-center cohorts. Additionally, not all growing nodules were

404  pathologically confirmed to be malignant. Hence, the criteria for surgical intervention after

405  nodule progression warrant further exploration.

406      In summary, developing individualized management strategies for SSNs is crucial in

407  clinical practice. Based on this study's distinctly different natural courses of SSNs, we

408  combined clinical and radiomic features using machine learning algorithms to create predictive

409  models for assessing whether SSNs will grow within two years. Compared to previous research,

410  our models demonstrated improved predictive performance. Additionally, for the first time, we

411  used multi-center, long-term follow-up data to establish a predictive model for SSN growth

412  within five years, guiding long-term follow-up of SSNs. These predictive models were further

413  validated in an external dataset, demonstrating good generalizability.

414

415  **CRediT authorship contribution statement**

416

417  **Declaration of Competing Interest**

418      The authors declare that no commercial or financial relationships that could be construed as

419     a potential conflict of interest existed during the research.

420     **Acknowledgments**

421     No acknowledgments.

422     **Funding**

423

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
