# OpenReview forum: "Development and Validation of an Explainable Machine Learning-Based Model for Predicting the Interval Growth of Pulmonary Subsolid Nodules: A Prospective Multicenter Cohort Study"
_KDD.org/2024/Workshop/AIDSH — KDD-AIDSH 2024 Poster_

### Official Review · Reviewer_pU3n · 2024-06-12
**No figure or table**

**Rating:** 4
**Confidence:** 1

**Review:**

First of all, this submitted paper is not blindly. And there's no figure or table attached in the submitted pdf, although the figure and table numbers were mentioned in the pdf. Not sure whether this is a system error or authors' incorrect operation. Basically, as there are no results, it's not easy to read the paper.
Pros:
* Well-structured:  this paper is well-organized with clear sections, making it easy to follow the research process.
* Comprehensive data analysis: data were collected from 3 medical centers with extensive data analysis, and validation processes are performed, ensuring the reliability of the findings.

Cons:
* No figures or tables.
* Incremental advancement: while innovative, the work might be seen as an incremental advancement rather than a breakthrough, building on existing models rather than introducing entirely new concepts.
* Implementation challenges: the transition from a predictive model to real-time clinical implementation might face practical challenges not fully addressed in the paper.

---

### Official Review · Reviewer_8RMD · 2024-06-18
**Reviews from 8RMD**

**Rating:** 4
**Confidence:** 4

**Review:**

This study developed and validated a machine learning-based CT radiomics model to predict the growth of pulmonary subsolid nodules (SSN) over time for personalized follow-up.Data from 642 patients with 717 SSNs were analyzed, and models were developed using optimal ML algorithms, with performance measured by AUC and feature importance ranked by SHAP.The models showed high accuracy in predicting SSN growth at 2 and 5 years, with AUCs up to 0.911 for internal and 0.734 for external validation, providing a basis for improved clinical management strategies.

The background, significance, data sources, and analysis of the paper are solid, but there is a lack of technical innovation. XGBoost, Light GBM, and SHAP are classical.

Serious issues:
1.	Violated the double-blind process (submissions will undergo a rigorous double-blind review process).

2.	Incorrect text format (Submissions to this KDD conference should consist of a main text of up to 5 pages using the ACM style file, excluding references, with unlimited space allowed for the appendix).

3.	Missing figures and tables.

---

### Decision · Program_Chairs · 2024-06-28

Accept (Poster)